# Nanoscale MOSFET as a Potential Room-Temperature Quantum Current Source

**DOI:** 10.3390/mi11040364

**Published:** 2020-03-31

**Authors:** Kin P. Cheung, Chen Wang, Jason P. Campbell

**Affiliations:** 1Nanoscale Device Characterization Division, National Institute of Standards & Technology, Gaithersburg, MD 20899, USA; jason.campbell@nist.gov; 2Currently State Key Laboratory of ASIC and System, School of Microelectronics, Fudan University, Shanghai 200433, China; wchen@dlut.edu.cn

**Keywords:** nanoscale, mosfet, quantum current

## Abstract

Nanoscale metal-oxide-semiconductor field-effect-transistors (MOSFETs) with only one defect at the interface can potentially become a single electron turnstile linking frequency and electronic charge to realize the elusive quantized current source. Charge pumping is often described as a process that ‘pumps’ one charge per driving period per defect. The precision needed to utilize this charge pumping mechanism as a quantized current source requires a rigorous demonstration of the basic charge pumping mechanism. Here we present experimental results on a single-defect MOSFET that shows that the one charge pumped per cycle mechanism is valid. This validity is also discussed through a variety of physical arguments that enrich the current understanding of charge pumping. The known sources of errors as well as potential sources of error are also discussed. The precision of such a process is sufficient to encourage further exploration of charge pumping based on quantum current sources.

## 1. Introduction

Accurate electrical measurements are the very foundation of modern science and the accurate measurement of electric current is particularly challenging due to the lack of a convenient standard. As part of the effort to bring an end to the use of physical objects to define measurement units, the 26th General Conference on Weights and Measures redefined the ampere as the flow of 1/(1.602176634 × 10^−19^) elementary charges per second [1], linking electric current to the fundamental unit of charge. Hailed as a landmark moment in scientific progress, this redefinition using a natural constant brings a new urgency to the development of a current source that adheres to such a definition. This is advantageous not only as a practical calibration standard but also to complete the quantum metrology triangle (QMT) for checking the consistency of fundamental constants [2]. 

Efforts to control the serial flow of electrons to define current have been ongoing for three decades, starting with the electron turnstile operating at millikelvin temperature [3,4,5,6,7]. These turnstile measurements realize current, *I*, such that *I = ef* where *e* is the elementary electronic charge and *f* is the control frequency. Such a source will meet the modern definition of current exactly. While there have been demonstrations meeting the metrology precision needs [7], the current level is limited to the picoampere range, which is too low for practical purposes. Various alternatives have been explored to increase the current level [8,9,10,11,12,13,14], however, the increased current level costs accuracy. While some of these approaches have shown theoretical pathways to achieve accuracy at higher current levels [15], the measurements require cryogenic operation temperature and high magnetic fields [16]. If realized they will not be practical current standards deployable outside of a highly specialized laboratory.

## 2. Materials and Methods 

All the single electron transistors (SET) or electron turnstiles rely on a Coulomb blockade process. Artificially fabricated quantum dots are not generally small enough and therefore cryogenic operation is a necessity. Note that a 2014 review of quantum current sources picked the tunable barrier pump as the leading candidate for a primary current standard because it requires “only” the pumped helium-3 cryostat instead of a dilution-refrigerator [17]. For the proposed quantum current source in this work, the quantum dot is replaced with a broken bond, and instead of a Coulomb blockade, the Pauli’s exclusion principle is relied upon to enforce the quantized flow of electrons. 

Production quality nanoscale metal-oxide-semiconductor field-effect-transistors (MOSFETs) are small enough that the interface between the gate dielectric and the channel can have only a few defects or no defect at all. MOSFETs with only one interface defect can be down selected from arrays of billions of MOSFETs to generate a statistically relevant population of single defect devices suitable for parallel charge pumping to achieve an accurate quantized current source at relevant current levels. 

The well-known charge-pumping (CP) process [18] can potentially meet this quantized current source need as it supposes to shuttle one charge from the source and drain of the MOSFET to the substrate per interface defect per charge-pumping cycle. This interpretation of CP has been widely accepted [19] but never fully demonstrated at the single defect level. A few reports in the literature [20,21,22,23,24] claimed to have observed one charge per cycle per defect, but there is always more than one charge per cycle in all these reports and the precision has been poor, until now. 

Commercial n-channel MOSFETs with 50 nm effective channel length and 100 nm channel width were measured using the charge-pumping technique described in [18] (switching between strong accumulation and strong inversion) (Figure 1). The gate oxide is 1.6-nm-thick SiON. The square wave gate voltage switched from −1.3 V to +1.25 V, satisfying the strong accumulation and strong inversion conditions while not completely overwhelming the measurement with gate leakage current. The 0.1 V source and drain bias is the experimentally determined lowest net-zero leakage point. Many devices were measured and devices with 0, 1, 2, and 3 (and more) defects were found and easily identified by the quantized nature of the measured current level. This is consistent with the expected interface defect density (10^10^/cm^2^) in commercial MOSFETs. Devices with only one defect provide the most unambiguous proof and are the focus of the following discussion.

## 3. Results

Figure 2 shows the measured charge-pumping current as a function of frequency (100 kHz to 40 MHz) with three different gate square wave rise and fall times, on the same transistor. The noise floor of the measurement set up is 10 fA root-mean-square. Since each data point is the average of 800 measurements, the uncertainty of the measured current is on the order of 0.3 fA. The accuracy of the frequency has not been checked, however. The frequency response is clearly a linear function that satisfies *I = nef* with *ne* the slope of the line in the unit of Coulombs per cycle. There are two lines for each rise/fall time. One is the raw (open diamond) data, and the other is gate leakage current corrected (solid circle) data.

To perform the gate leakage correction, we begin with the knowledge that charge-pumping current at zero frequency should be zero. Thus, the measured net current at zero frequency is entirely due to gate leakage. Since gate leakage current is an exponential function of gate voltage, gate leakage primarily occurs during the strong accumulation/inversion portions of the gate waveform and is small during the rising and falling edges of the gate waveform. For fixed rising/falling edges, increasing the frequency of the gate waveform decreases the time spent in strong accumulation/inversion and decreases the effective gate leakage current [25]. A common problem in the charge-pumping experiment is that when the frequency is too low, the measured current is dominated by leakage current and noise, providing little information of value. Thus, most experiments start at a moderate frequency, which in our case is 100 kHz. As seen in Figure 2, this frequency is low enough that in the linear plot it is very close to the zero-frequency point. This leads us to approximate the measured current at 100 kHz as pure leakage current *L_leak_*. The frequency-dependent gate leakage can be calculated (assuming an ideal trapezoidal waveform) from the known rise/fall time *t_r_*. The method is as follows: At each frequency *f*, the period is *τ* = 1/*f*. Subtracting the rise and fall time from this time period results in the time period during which gate leakage occurs. Thus, the leakage contribution to the measured current, *A*, decreases with increasing frequency by a fraction of *R =* 2*t_r_/τ.* The corrected charge-pumping current, for each frequency, is therefore CP = A − (1 − R)L_leak_.

After subtracting leakage current from the measured current, the slopes of the post correction data are steeper. Converting the slope to charge per cycle using *e* = 1.6022 × 10^−19^ Coulomb, we get *n* = 0.996, 0.967 and 0.955 for the rise/fall time of 8 ns, 4 ns and 2.5 ns respectively. These *n* values are closer to 1 than any reported in the literature and they strongly suggest that one charge is indeed pumped per defect per cycle. There is an apparent trend of a bigger deviation to one charge per cycle as rise/fall time gets shorter. It is most likely due to increases in the deviation (overshoot and ringing) to perfect trapezoidal waveform with faster rise/fall time. Such deviations make leakage correction less accurate.

Note that a traditional explanation of *n* dependent on rise/fall time predicts an opposite trend—*n* increases with faster rise/fall time [26]. This explanation is based on the idea of emission loss. The assumption behind the emission loss mechanism is a continuous distribution of defect energy states spanning the entire silicon band gap. This explanation clearly cannot work when there is only one defect.

## 4. Discussion

The results shown here are clearly confounded by the leakage current, and the correction procedure is somewhat limited. Notice that the leakage-corrected data for the 2.5 ns rise time plot in Figure 2 is missing the first point, which has a value of −8.9 fA, suggesting a slight over-correction exists. There are two sources of leakage, namely gate leakage and junction leakage. Gate leakage can be eliminated by simply using a thicker gate oxide. For example, increasing the gate oxide thickness from 2 nm to 7 nm will reduce the gate leakage by as much as 15 orders of magnitude [27]. For junction leakage, a properly made junction can have leakage down to less than 10^−19^ A for the small device geometry used in this work [28]. Fortunately, these very properties are also required for the cell transistor of dynamic random-access memory (DRAM), and DRAM is a very well-developed technology. Thus, testing the method on DRAM cell transistors is the natural next step.

It is useful to examine if the elimination of leakage could lead to high enough accuracy. 

Figure 3 shows the charge-pumping concept in detail: At strong inversion (Figure 3A), the surface potential barrier in the channel (blue curve) is suppressed so that electrons from source and drain flood the channel (red arrows), allowing the empty defect site (x) to capture an electron (red circle). As the gate voltage is reversed toward depletion, the surface potential barrier increases and electrons from the channel flow back out to source and drain as shown in Figure 3B. The electron that was captured by the defect remains as shown in (Figure 3C). As the gate voltage continues to change toward strong accumulation (Figure 3E), holes from the substrate flood the channel. The captured electron at the defect site can recombine with a hole (the red circle disappears). As the gate voltage changes back toward depletion, all the holes flow back to the substrate, except the one that recombined with the electron. The empty defect is now ready to capture an electron the next time the channel is flooded with electrons (Figure 3A) and the net result of one charge-pumping cycle is to move one electron from the source and drain to the substrate (as a missing hole).

Provided that the gate leakage can be minimized, the crucial remaining barriers to utilizing charge pumping to achieve a high precision quantized current source are insurance that: (1) there is only one charge captured per defect, (2) the capture process is fully complete, and (3) the captured charge will not be lost by emission before recombination occurs.

Perhaps the most important question surrounding this approach is an insurance of single charge capture per defect site per cycle. It has been shown that only interface defects contribute to the charge-pumping current [29]. It has been argued that the interface defect is a *P_b0_* center [30,31], which is an interfacial dangling bond on a silicon atom, which is back bonded to three additional silicon atoms. This silicon dangling bond defect is well-known to be amphoteric, meaning that it can capture an electron to become negatively charged or lose an electron to become positively charged, as suggested by electron spin resonance (ESR) spectroscopy [30,31]. This would mean Figure 3 is incomplete. While the process represented in Figure 3G to Figure 3A involves the neutral dangling bond defect capturing an electron to become negatively charged, the neutral dangling bond losing a charge is omitted as it should happen between Figure 3E and Figure 3F. However, this would mean two charges are pumped per CP cycle instead of one and would be inconsistent with the data in Figure 2.

The idea that all true interface states are *P_b0_* centers is not universally accepted [32,33]. There could be another true interface state that is not amphoteric. If the defect responsible for results shown in Figure 2 is one such non-amphoteric defect, the mystery is solved. Though, this is unsupported by the majority of the ESR literature. 

If the defect is a *P_b0_* center, one possible reason for the discrepancy is that the defect can still be amphoteric but only one of the charged states is involved in the CP process. The evidence of the amphoteric nature of the interface states comes mainly from capacitance-voltage (CV) and electron-spin-resonance (ESR) studies [30]. There are clearly two defect energy peaks in some measured CV curves separated by ~0.7 eV, indicating that the defect energy level is gate bias-dependent but has two stable values. Similarly, the ESR signal remains strong between the energies defined by these two CV peak locations, which clearly suggests that both peaks represent the neutral silicon dangling bond. At energies beyond these two peaks, either higher or lower, the ESR signal vanishes, indicating the defect has either captured an electron to become negatively charged or lost the electron to become positively charged [34].

Here we propose that the two CV peaks correspond to two stable configurations of the same dangling bond under the bias. One might expect, when the gate bias is in inversion, the silicon dangling bond has more sp^3^-like character (tetrahedral, the field vertical to the interface pulls the electron away from the silicon atom) and resembles the expected configuration of a defect with a captured electron. This configuration is also called the acceptor state. Similarly, when gate is negative, the same silicon atom has a more sp^2^-like character (planar, the field pushes the electron toward the atom) and resembles the configuration of the positively charged defect (the silicon atom has three bonds, or six bonding electrons). This configuration is also called the donor state [34].

Note that silicon normally does not bond in sp^2^ fashion because the remaining p_z_ orbital must form a π bond [35], which is difficult due to its size. However, for the donor state to be an efficient configuration to capture a hole, it must resemble the configuration of the final state—missing an electron, or a silicon atom that has only three valence electrons. Since the fourth electron is absent, there is no need to form the π bond when the rest of the three electrons form sigma bonds with neighboring silicon atoms. In the intermediary state, before hole capture, the fourth electron is localized on the silicon but sp^2^ reconfiguration has not yet occurred due to the energetic cost. The fourth electron remains non-bonding (dangling), and there is only a small energy penalty to lose this non-bonding electron. (capture a hole if available).

As sp^3^ and sp^2^ are known stable configurations of hybridized orbitals, intermediary configurations are symmetry forbidden and therefore unlikely. At energies between these two peak values, we do not have other configurations but a distribution between these two configurations. The total number of dangling bonds remains unchanged. This explains why the ESR signal remains flat.

Between stable configurations (low energy levels) is an energy barrier, making the conversion between one configuration to another only observable when the gate voltage is swept very slowly (as those in ESR or CV experiments). To get an idea of how hard it is to convert between these two configurations, an analogue may be drawn from reconfiguration energy in the capture and emission charge in the near interface dielectric region. It has recently been shown that this reconfiguration energy is of the order 0.7 eV, and capture is followed by an extremely slow emission due at least in part to the large configuration energy barrier associated with capture [36]. Since the donor/accepter configurations discussed above are also subject to a strong driving force—the electric field, the sp^2^/sp^3^ reconfiguration is observable when the voltage sweep is very slow. The situation is quite different in CP experiments where the time spent at strong inversion or accumulation is too short for the neutral dangling bond defect to change from one configuration to the other even though the field is switched back and forth. In other words, while both dangling bond configurations are accessible in CP experiments, only one configuration participates in the charge pumping process. Figure 4 illustrates the two possible CP cycles each involving only one configuration of the same dangling bond defect.

Figure 4a,b shows how CP starting with one energy state of the dangling bond will stick to that same energy state throughout the CP cycle. In both cases, panel c illustrates that capturing a hole by the accepter state or capturing an electron by the donor state is prohibited (large energy barrier) without reconfiguration, which occurs on time scales much longer than the CP half cycle. Note that in both cases, the CP cycle can start at panel c instead of a, and the result is equivalent. This explains the data in Figure 2 and why there is only one charge per interface defect per CP cycle. We note that this speculative explanation of the charge pumping process does deviate from the conventional explanation. However, this fine point may have been lost in the initial formulations of the CP process since the early observations were undoubtedly dealing with an energy continuum of interface states due to the larger defect densities. In the limit of charge pumping a single interface state defect, the charge capture process deals with discrete energy levels that act to further confound CP interpretation. We note that this already complex scenario can be further complicated by the presence of the *P_b1_* interface defect variants [37,38].

Figure 4 also illustrates several factors that can potentially degrade the precision of the quantum current source. As per the above discussion, it is somewhat difficult to ensure that only one charge is pumped per defect per cycle. From the above discussion, the barrier that prohibits an accepter state to act like a donor state or vice versa is not infinite. One might be able to modify this barrier by modifying the strength of the accumulation or inversion biases as well as the time available to overcome it (CP frequency). Limiting the accumulation and inversion biases and increasing the CP frequency obviously are fruitful propositions. However, these measures will worsen our insurance that the charge capture process has enough time to complete (factor 2).

Referring to Figure 4, panels a and c for both cases L and R represents the capturing processes. The completeness of charge capture (factor 2) depends on the capture rate and the available capture time (half the CP period). The capture rate is dependent on the density of carriers available. Thus, the time available for capture and the density of available carriers are the keys to this obstacle. Fortunately, the density of carriers available for capture, and thus the capture rate, grows exponentially with increasing inversion or accumulation bias. So, to achieve complete capture, the inversion and accumulation biases need to be increased and the capture time needs to be extended. We note that both requirements are the exact opposite tactics used to ensure that only one charge is captured per defect per cycle (factor 1). An optimum compromise exists and that will determine the ultimate precision of the quantum current source.

Figure 4 panel b of both cases L and R illustrates the process that is susceptible to emission loss (factor 3). As the gate voltage is swept from strong inversion/accumulation through depletion toward accumulation/inversion, there is a short time window when the captured charge can emit out of the defect state. This time window is a small fraction of the rise/fall time of the CP waveform, and the probability of emission depends on the depth of the defect energy state in the silicon band gap. The traditional CP theory [26] predicts finite emission loss regardless of how fast one makes the rise/fall time because it assumes a continuous distribution of defect energy states throughout the silicon band gap. As mentioned above, this picture clearly not applicable here. The depth of both *E_t1_* and *E_t2_* are known and they are deep enough that they would be normally considered negligible in traditional CP. However, using CP as a high precision quantized current standard requires a consideration of this emission loss. Again, shorter CP rise/fall time is the direct means to minimize this factor.

## 5. Conclusions

In summary, we propose and demonstrate a potential quantized current source based on charge pumping in a nanoscale MOSFET. Confounding factors such as gate leakage, single and complete charge capture per cycle, and emission loss have been discussed in detail. Many of the experimental tradeoffs discussed above are in conflict such that an experimental middle ground must be explored before one can estimate accuracy. The potential of using CP as a quantized current source is clearly demonstrated. The advantage of room temperature operation, potential for high current level, and the reliance on matured technology make the proposed method quite appealing as a quantized current standard that can be deployed outside the laboratory.

## Figures and Tables

**Figure 1 micromachines-11-00364-f001:**
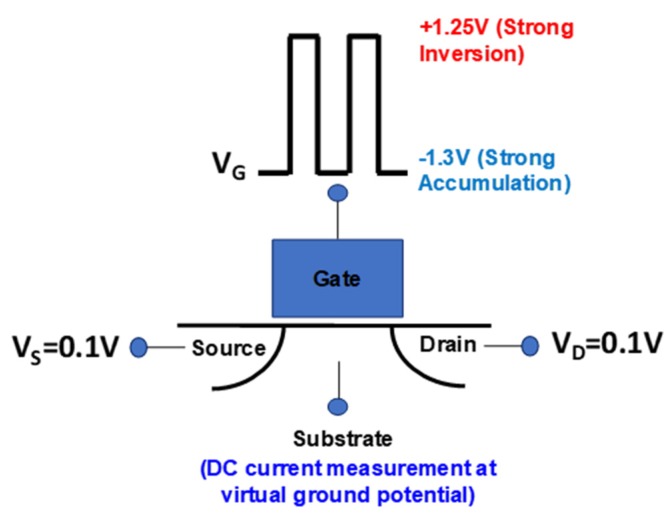
MOS charge-pumping measurement arrangement. Square wave is applied to the gate to switch the channel between strong accumulation and strong inversion. Source and drain are set to 0.1 V. Charge-pumping current is measured from the substrate as DC using a current amplifier with virtual ground input.

**Figure 2 micromachines-11-00364-f002:**
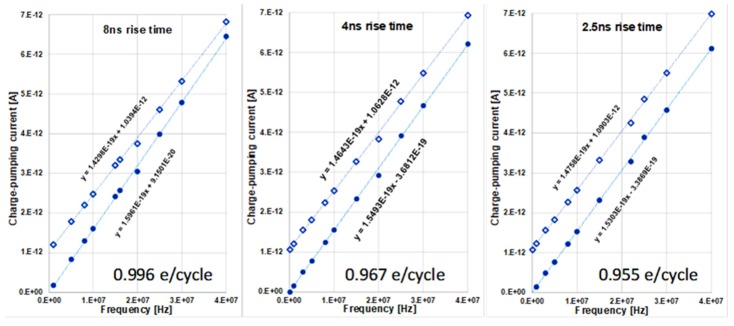
Measured and leakage corrected charge-pumping current as a function of frequency and their corresponding least square fits for three different rise/fall times. The charge per cycle at the bottom is extracted from the slope of the leakage corrected least square fit line, dividing by 1.6022 × 10^−19^ Coulomb.

**Figure 3 micromachines-11-00364-f003:**
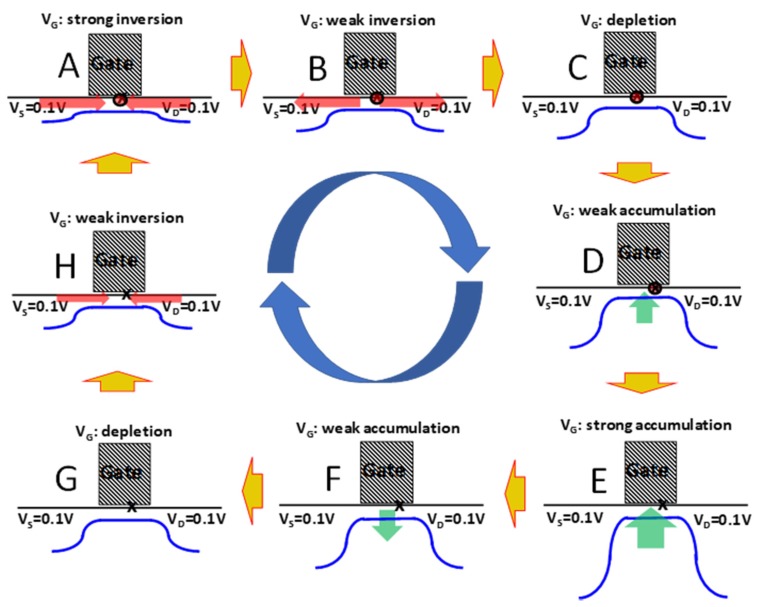
Illustration of the charge-pumping concept. Blue line is the surface potential drawing in one dimension varying from source to drain. Red arrows indicate electron flux (minority carriers in a n-channel metal-oxide-semiconductor field-effect-transistor (MOSFET)). Green arrows indicate hole flux. X indicates an interface defect, and x with a red circle indicates an electron is captured on the defect site. The cycling of the gate voltage from strong inversion (**A**) to depletion (**D**) to strong accumulation (**E**) to depletion (**G**) and back to (**A**) serves to modulate the surface potential in the channel and therefore controls the flow of electrons from source and drain and holes from the substrate.

**Figure 4 micromachines-11-00364-f004:**
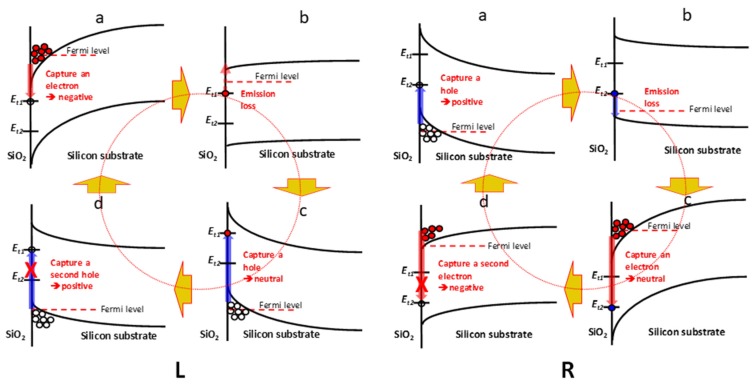
Band diagram of the Si/SiO_2_ interface showing the two energy levels, *E_t1_* and *E_t2_*, of a dangling bond defect. **L**: CP cycle involving only the sp^3^-like accepter energy state *E_t1_*. (**a**) At strong inversion, the *E_t1_* state captures an electron to become negatively charged; (**b**) sweeping through depletion toward accumulation provides a small time-energy window for emission loss; (**c**) at strong accumulation, the *E_t1_* state captures a hole to become neutral again; (**d**) while still accumulated, the neutral *E_t1_* state does not have enough time to convert to *E_t2_* state and cannot capture another hole to become positively charged. **R**: CP cycle involving only the sp^2^-like donor energy state *E_t2_*. (**a**) At strong accumulation, the *E_t2_* state captures a hole to become positively charged; (**b**) sweeping through depletion toward inversion provides a small time-energy window for emission loss; (**c**) at strong inversion, the *E_t2_* state captures an electron to become neutral again; (**d**) while still inverted, the neutral *E_t2_* state does not have enough time to convert to *E_t1_* state and cannot capture another electron to become negatively charged.

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
