# Peer review of "Nanoscale MOSFET as a Potential Room-Temperature Quantum Current Source"

_micromachines, 2020, doi:10.3390/mi11040364_

Round 1

Reviewer 1 Report

The authors have performed an interesting study on quantized current source using a nanoscale MOSFET. Even though the work is interesting, certain aspects of the manuscript needs to be improved before the manuscript can be suitable for publication.

  1. One of the issues of this work is that it relies on manufacturing defects which may or may not occur (and potentially may be uncontrolled to an extent).
  2. Another potential issue could be the response time of the trap and generation of other defects under strong accumulation/strong inversion due to higher electric field in the oxide.
  3. Add reference for commercial MOSFETs in line 64
  4. What is the measurement noise floor in fig. 2.
  5. Please add a discussion with respect to the nature of the trap (its time constant etc.).
  6. How many cycles before stress causes additional defects? Is this technique robust?

Author Response

We thank the reviewer for insightful comments. They are addressed point by point below:

1. One of the issues of this work is that it relies on manufacturing defects which may or may not occur (and potentially may be uncontrolled to an extent).

Lines 51 to 56 in the manuscript discussed this point. This is why it has to be a nano-scale transistor. It is true that single defect may or may not occur. However, statistics dictates that a large percentage of these transistors will have zero defect and a large percentage will have one defect. It is not hard to produce millions of transistors on a chip and hook them up in on-chip circuits so that those with one defect will be mapped out and only these devices will be used in single device or multiple devices in parallel based current standard. We already discussed this in the manuscript and mentioned that down selection is required (with on-chip measurement capability, of course.)

 2. Another potential issue could be the response time of the trap and generation of other defects under strong accumulation/strong inversion due to higher electric field in the oxide.

Response time is indeed an important factor. We actually discussed this in detail (lines 247 to 256). As for new trap generation during use, this is of course an issue that must be characterized carefully in future works. Sooner of later, new trap will form. The way around this is to have an on-board scanning (or re-map) to ensure that only the single defect transistors are used. One can perform a periodic self-calibration to do this. Of course, when the oxide field is very high, trap generation can become rapid. However, decades of oxide reliability literature suggest that at the moderate field needed, this should not be a problem.

3. Add reference for commercial MOSFETs in line 64

We agree that the proper way is to identify the source of the device. However, we have to apologize for not doing so because we are bounded by confidentiality and cannot identify the source. It is from one of the largest foundries.

4. What is the measurement noise floor in fig. 2.

Good catch! We added discussion on this question (line 79 to 82). Our measurement noise if floor is very low: 0.3fA. So the error bar is smaller than the size of the data point itself.

5. Please add a discussion with respect to the nature of the trap (its time constant etc.).

Arguably, line 155 to line 268 are devoted to address this very question. We are not certain what the reviewer mean by add a discussion. We have, in previous publications (cited), established that only true interface states will take part in charge-pumping. It is generally accepted that silicon dangling bond (Pb center) is the only true interface state at the SiO2/Si interface. We even mentioned the rare dissenting view with reference in our discussion.

Capture time constant for interface states varies by 22 order of magnitude in the literature depending on the measurement method used. This is something to be studied in the future. The fact that we do not see any deviation from a straight line even up to 40MHz suggest the capture time constant is very short under our measurement condition. How far can we push the frequency is something we like to study.

6. How many cycles before stress causes additional defects? Is this technique robust?

As we discussed in our respond to comment number 2, decades of gate oxide reliability studies suggest that under moderate oxide field (one that will be use) new defect will form eventually. However, it is not a fast process. When the transistor is very small, even adding one interface defect will shift the threshold voltage high enough to turn the transistor into a failed device. As typical device lifetime is 10 years, it is reasonable to expect that the transistor will operate for years before additional defect will occur under normal operation condition. Under the mildly accelerated condition (stronger accumulation and inversion), this will be shorter. It is reasonable to expect the device will operate for months running continuously before the addition of the next defect. Thus, we expect this technique to be robust (with self-calibration once every few months)

Reviewer 2 Report

Interesting work on CP in small area MOSFET.

It could deserve publication after major revision.

The fact to show Icp vs frequency curve for a single Vglow Vgup voltages is quite limuited in the analysis of CP.

The authors must include the conventional Elliot plot i.e. Ic fo given ferquency but varying Vgabse level with constant Vglow-Vgup amplitude to clearly eveidence the single defect analysis.

Some statitics of results should also be given since defect number in small area devices is variable from deviec to device. Only one set of curves of Icp vs f on one sample is not enough.

Author Response

The reviewer demands that we measure the Elliot plot to prove that there is indeed only one defect state without explaining why. In general, one performs more measurement when the existing one is not sufficient to support the conclusion. In this case, what other possibility can there be? It cannot be zero defect state because there would not be any CP current. Can it be two defect states, leading to exactly 0.5 electron per CP cycle? Not possible! Can it be many states each with very low fractional charge being pumped per cycle? Even more crazy. With no other possible/alternative explanation, we have met the sufficient criteria and adding more measurement is simply a waste of readers' time. Therefore, we disagree with the reviewer that measuring the Elliot plot is needed.

It is true that the number of defects per device varies significantly in small devices. We discussed that in the text already. We did measure a number of devices and experimentally observed other number of defects per device, including 2, 3, 4 5,.. We also mentioned this in our text already. We explained, in our text that the one defect case offered the best demonstration of the idea and focus on it. In our view, adding the results from those other measurement do not add to the paper and will only serve to waste the readers' time. We understand the desire to know how much variation exist even for all the one defect only devices. However, as discussed in the manuscript, the devices are far from perfect due to significant level of leakage and the correction is not perfect. As such, examining the variation (how close to being 1 charge per cycle) is not meaningful at this point. As long as one cannot argue that the result can simply be a happy coincident, we believe it is enough as the first demonstration.

Base on the above, we did not make any modification of the manuscript in respond to this reviewer's comment.

Reviewer 3 Report

The idea of combining the charge pumping technique with a one-defect MOSFET to obtain a quantized current source deserves publication. Experimental results in Figure 2 are of utmost importance to demonstrate the usefulness of this technique. However, there are some points that are not fully clear for this reviewer:

provide more detail on the calculations and on the measured leakage current used to obtain the corrected curve provide a frequency range where your curves will keep being linear (I guess that larger frequencies will lose linearity)

Author Response

We thank the reviewer for pointing out the need for more detailed discussion on how the leakage correction is done. Accordingly, we modified that part of the manuscript extensively to provide the needed details.

We agree with the reviewer that as frequency goes up, deviation from linearity will occur at some point. The data shows that it hasn't occur at 40MHz. Unfortunately, we do not has data at higher frequency to define the range of linearity. On the other hand, even if we did, the result is not going to be applicable to other devices with different designs. In other words, we expect the linear range can be optimized by device design.

Round 2

Reviewer 1 Report

The authors have answered my questions satisfactorily.

Reviewer 2 Report

If you do not want make any change, I cannot agree on that position. So I let decide the editor...

Author Response

We agree to disagree.

Reviewer 3 Report

Reviewer's questions have been satisfactorily answered.

Author Response

Thank you.